# Measuring the Risk of Violence Through Health Surveillance

**DOI:** 10.3390/ijerph21121708

**Published:** 2024-12-22

**Authors:** Nicola Magnavita, Igor Meraglia, Giacomo Viti, Martina Gasbarri

**Affiliations:** 1Occupational Epidemiology Unit, Department of Life Sciences and Public Health, Università Cattolica del Sacro Cuore, 00168 Roma, Italy; igor.meraglia01@icatt.it (I.M.); giacomo.viti01@icatt.it (G.V.); 2Health Surveillance Service, Local Healthcare Unit Roma4, 00053 Civitavecchia, Italy; martina.gasbarri@aslroma4.it

**Keywords:** aggression, harassment, methodology, occupational epidemiology, participatory methods, prevention, risk assessment, risk management, threat, workplace

## Abstract

Workplace violence (WV) is a ubiquitous, yet under-reported and under-studied phenomenon. Prevention measures may be ineffective because risk assessment is often based on unvalidated algorithms. After monitoring the risk of WV in a healthcare company for over 20 years, this paper presents the results collected in 2023 and details of the methodology used. Monitoring WV in health surveillance can involve three actions: (1) asking all the workers who attend periodic medical examinations in the workplace whether they have suffered physical aggression, threats, or harassment in the previous year; (2) investigating WV at the end of workplace inspections by setting up participatory ergonomics groups (PEGs) to suggest solutions; (3) investigating the characteristics and consequences of WV through anonymous online questionnaires. In 2023, 6.9% of the healthcare workers (HCWs) reported having experienced one or more physical attacks during the previous year; 12.7% reported having been threatened, and 12.9% reported other types of violent, harassing behavior. The HCWs observed an increase in violence after the pandemic years and, in the PEGs, suggested using different preventive measures in different health departments. The online survey provided further information on the characteristics of WV and its consequences. The risk of WV can be effectively characterized and measured through health surveillance.

## 1. Introduction

Man is a violent animal. Aggression, which may have contributed to the spread of the species and its supremacy over other animals, has an important phylogenetic element [1]. Aggressive behavior in humans is of two kinds: proactive aggression motivated by low emotionality and high levels of instrumentality, used in order to acquire benefits, and reactive aggression, which is characterized by emotional lability, or the propensity to behave impulsively in response to provocation. The two subcategories of violence correspond to separate psychobiological patterns with underlying genetic [2], neuro-psychophysiological [3,4,5,6], metabolomic [7,8,9], and hormonal [10] markers [11]. Compared to many other primates, humans have a high propensity for proactive aggression, while they have a low propensity for reactive aggression [12].

The characteristics of human beings make workplace violence (WV) an unavoidable hazard. However, like all occupational risk factors, it can be measured, predicted, controlled, and reduced through a risk management process. WV must be prevented at all costs because, in addition to harming the victims, it interferes with production which is the purpose of work activity [13,14,15,16,17,18]. Although everyone is aware of this principle, WV is widespread in all sectors where it represents a risk that is often unacknowledged and poorly studied [19,20,21]. Scientific research has long neglected this topic. The idea that violence in the workplace needed to be addressed did not appear in scientific literature until the late 1980s, when the first studies on the role of violence as an occupational stressor [22] and the need to assist victims of violent trauma were published [23,24]. Only in the mid-1990s did it become clear that the economic damage caused by WV is quantified not only by the number of sick days suffered by the HCWs (fortunately a limited number) who suffer the greatest physical damage, but also by the considerable emotional and motivational effects of the various forms of violence [25]. In the following years, Convention No. 190 on Violence and Harassment [26] and Recommendation No. 206 [27] of the International Labor Organization (ILO), the Framework Guidelines for Addressing Workplace Violence in the Health Sector of the World Health Organization [28], and the European Agency for Safety and Health at Work [29] stressed the importance of WV and the need to prevent it. However, at the present time, only 32 nations have ratified the International Labor Organization Convention against Violence and Harassment, and only 20 of those countries, including Italy, have put it into effect. Italy is one of the few countries in the world that considers violence to be an occupational risk and requires the employer to evaluate it and prepare all the necessary prevention and protection measures [30].

Violence can be classified as physical, verbal or psychological. Examples of psychological violence include sexual harassment [31], bullying [32,33], and incivility [34]. In the occupational environment, verbal abuse has been shown to be the most common form of violence [35,36], but also the one that can have the worst consequences, especially for young people undergoing training [37]. WV can also be classified according to the type of perpetrator. Type I refers to acts of violence, usually with criminal intent, carried out by those who have no legitimate occupational relationship with the organization or its employees. These are events generally associated with crimes such as theft, robbery, or terrorism [38]. Type II refers to violence perpetrated by clients, healthcare patients, or their visitors and relatives [39,40,41]. Type III refers to violence performed by colleagues or company staff, [42]. Type IV is violence carried out by a person outside the company who has a personal relationship with an employee [43,44].

Workers in healthcare and personal assistance services are mainly exposed to Type II events [45], although the possibility of other types of violence cannot be excluded. In the United States, studies have demonstrated that although HCWs make up only 13% of the US labor force, 60% of workplace assaults occur against them and WV is the second most common cause of fatal workplace injuries [44]. WV in healthcare is particularly serious because, in addition to attacking a caregiving profession, it ends up involving the quality of care and the health of the patient. In 2007, the Italian Ministry of Health issued a Directive to national health service facilities obliging them to monitor workplace violence [46]. Even in countries where employers are not obliged to prevent the risk of workplace violence, health companies are active in managing this risk, because violence is one of the parameters the Joint Commission monitors in order to evaluate the quality of health care [47].

The prevention of occupational risks in European countries is based on risk assessment. The evaluation process consists of different stages involving the identification of the problem, the measurement of the risk, and the definition of its characteristics. The control and monitoring measures implemented require final verification of post-intervention working conditions. Information gathered on violence leads us to believe that a WV hazard can easily be identified in all work environments. However, the next step, involving the measurement and calculation of the risk, presents numerous pitfalls, since subjectivity plays an important role in violence reporting [48]. The underreporting of violent events is a serious problem because a lack of detection prevents a correct assessment of the risk and the preparation of effective measures [49,50]. Workers do not always report violence, either because they believe that due to its frequent occurrence, it should be considered part of the job [45,51,52,53], or because they have found that reporting does not lead to a reduction in risk or the punishment of those responsible [54,55,56]. According to Sjoberg et al., HCWs regard violence as an integral aspect of caregiving, although they simultaneously identify themselves as victims of this violence [57]. In addition to these individual factors, differences in the procedures used to collect reports also have a direct influence on the outcome: some companies rely on workers to complete incident reports electronically, while others assign managers or security personnel to data collection [58,59].

WV can be illustrated as a pyramid (Figure 1). The top of the pyramid consists of a few major events that cause serious physical or mental injuries requiring sick leave and that are, therefore, formally reported to superiors, the injury insurance company, and sometimes the police. A much larger number of events have no obvious physical consequences and can be kept quiet, or are reported only if the victims are asked an explicit question by those investigating WV. Although these events may have caused the victim some minor injury that did not lead to days of absence, or did not result in evident injury, they are nonetheless important for their effect on mental health. Studies demonstrated that even witnessing WV may be negative for the mental balance of workers [60,61,62].

Collecting formal reports of assault usually yields very low prevalence rates. For example, between 2010 and 2014, an Italian public health company, situated adjacent to the one we studied, recorded an annual rate of 0.24% assaults with sick leave for its 1800 employees. These assaults accounted for 7% of all workplace injuries [63]. In the largest pediatric hospital in Europe, located in the same geographical area as the public institution we studied, only 21 cases of physical aggression were recorded in 34 months (2019–2021) towards about 2800 workers, corresponding to a 0.25% annual rate [64]. These rates are obviously just the tip of the iceberg, so some public health companies have introduced strategies to increase reporting; for example, by increasing executive support, training, and worker participation [65], or by training workers to immediately report WV cases via electronic applications. This has resulted in increases of over 10 times the number of events reported [66]. In this way, the AOU University Hospitals of Turin, a complex with approximately 10,000 workers, recorded between 2015 and 2017 a rate of WV of 1.92% per year, with 3.3% of workers having experienced one or more assaults in that period [67]. From 2016 to 2020, the University of Insubria, with the hospitals of Como and Varese, observed considerable underreporting, with annual rates as low as 0.52%, while in 2021–2022, after the introduction of a standardized reporting system, annual rates rose to 1.67% (one assault), 0.41% (repeated assaults), and 2.08% (total) [68].

In contrast, ‘ad hoc’ investigations prompt workers to report all incidents of assault, even minor ones, and achieve much higher rates. Thus, for example, in the hospitals of Modena and Reggio Emilia, 96% of the emergency room nurses in charge of triage declared that they had experienced at least one episode of violence in the previous year [69]. In Merano, a prospective study conducted over a period of 20 months in 2022–23, recorded 91 attacks on 38 nurses and 11 doctors in the emergency room [70].

Systematic reviews with meta-analyses are mainly based on ‘ad hoc’ studies and, therefore, record higher rates than those derived from formal reports. A meta-analysis of more than 250 studies, carried out before the COVID-19 pandemic on over 330,000 individuals, revealed a very high risk for healthcare workers in psychiatric and emergency settings, especially in Asian and North American countries, although there were considerable differences in working hours and workplaces [71]. According to this meta-analysis, two-thirds of workers reported having experienced exposure to some form of WV in the previous year; 24.4% of cases involved physical violence, and 42.5% involved non-physical violence. In over two-thirds of cases, verbal abuse was the most common form of non-physical violence; this was followed by threats in one-third of cases, and sexual harassment in the remaining 12.4% [71]. The incidence of violence among Chinese nurses was 71% in one year [72].

During and after the pandemic, studies on WV against HCWs became more numerous. Currently, the pooled prevalence rate of physical violence against HCWs is estimated to be between 14% [73], 17% [74], 19% [75], and 23% per year [76], with values as high as 36% in working environments such as the emergency services [77]. An umbrella review of meta-analyses [78] estimated that the average rate of physical violence against HCWs is 20.8%. Non-physical violence also has high prevalence rates. For example, Varghese et al. estimated that 30% of HCWs receive threats annually [76], and even higher risks have been estimated by Liu et al. (33.2%) [71] and Lu et al. (39.4%) [73]. Undoubtedly, the pandemic put a strain on healthcare systems around the world and this has definitely impacted also the rate of WV to which HCWs have been exposed. When evaluating WV, the chronological criterion must be taken into due account. In the first phase of the pandemic, there was a sharp decline in episodes of violence, but this was followed by a gradual recovery towards pre-pandemic levels [79].

The data we have briefly presented are sufficient to demonstrate the importance and spread of the WV problem and the need to adopt stringent epidemiological criteria to evaluate its frequency and characteristics. Unfortunately, not all healthcare companies have efficient epidemiological services or the competency that enables them to point research in the most effective direction for protecting workers. A manager who intends to evaluate the risk has a series of available sources that include directly reported events, accidents at work, and claims for compensation for damages, but these sources are often sterile. Evaluators, therefore, tend to make up for the lack of data by collecting information from time to time during inspections and internal audits and using it to make a personal assessment of the frequency and severity of the episodes. This is then used to create non-validated algorithms that enable the risk to be classified as “high”, “medium”, or “low”. Regardless of the rather insignificant result, the entire procedure is counter-productive and should be replaced by an evaluation based on real data [80]. The creation of gendered, culturally-based, non-discriminatory, participatory, and methodical approaches for addressing violence-related issues is one of the recommendations for managing workplace violence [81]. For this reason, we are convinced that assessing WV risk can benefit from health surveillance activities.

For over twenty years, WV has been measured during healthcare activities in all the companies we monitor. Having demonstrated the sustainability, cost-effectiveness and efficiency of this risk assessment method [82], we aim in this work to indicate its characteristics, advantages, and limitations. To provide a practical illustration of this method, we took as an example the data collected in a healthcare company in 2023.

## 2. Materials and Methods

### 2.1. Population

In Italy, in accordance with the European Directive on Health and Safety at Work [83], workers exposed to occupational risks must undergo health surveillance by an occupational physician. A very high percentage of workers in all sectors are considered to be exposed to occupational risks and consequently subject to medical examination in the workplace; in Italy, more than 10 million medical examinations are carried out annually. The Consolidated Law on Health and Safety at Work [84] requires the physician to carry out inspections and medical examinations in the workplace and to inform workers of occupational risks and the results of health surveillance. Traditionally, we have always integrated risk prevention activities (which are mandatory by law) with health promotion intervention so that these are given the same annual cadence as prevention. Although participation in health promotion projects is not mandatory, generally over 85% of workers willingly take part in these activities [85]. To exemplify the application of this method, in this article, we have reported the results of the surveillance conducted in 2023 in a public health company in the Latium region of Italy. The authors of this article were part of the surveillance team, two as specialists and two as interns. Workers exposed to occupational risks were doctors, nurses, technicians, employees, and other categories that collaborate in healthcare activities. As in all Italian public healthcare, the population was predominantly female (about 70%), and the average age was high (over 46 years) [86].

Although health surveillance activities are mandatory by law and, therefore, do not require prior authorization, in accordance with the Personal Data Protection Code (Law Decree 30 June 2003 n.196), workers undergoing medical examinations were invited to give their consent to the management and electronic processing of their personal data by the occupational physician for statistical or scientific purposes, even after the end of the health surveillance period. By signing the personal health document, the workers also agreed to collective anonymous publication in accordance with the aforementioned Code, the principles of the ICOH Code of Ethics for Occupational Health operators [87], and Occupational Medicine confidentiality principles (Legislative Decree 9 April 2008 n. 81 [84]). Furthermore, the research project was approved by the Ethics Committee of the Università Cattolica del Sacro Cuore (protocol code 2896, date of approval 5 December 2019). Informed consent was obtained from all workers.

### 2.2. Measures

During health surveillance, the occupational physician measures WV by carrying out three activities: regular medical examinations, inspections, and surveys. The first activity is the simplest to perform and was started in the company in 2005. During their regular examination, the occupational health physician systematically asks workers whether, in the previous year, they had suffered a physical attack, a threat, or harassment in the workplace. For this purpose, we use the first three questions of Arnetz’s “Violent Incident Form” (VIF) questionnaire [88], a tool specifically designed to record violent events in healthcare activities. The VIF is based on an operational construct in which “physical assault” is defined as an attack, with or without weapons, that could cause or not cause physical damage; “threat” is defined as the intention to cause physical damage; and “harassment” is any annoying or unpleasant act (words, attitudes, actions) that creates a hostile work environment.

The second activity is introduced after completing the workplace medical examinations that the occupational physician is legally required to perform. After the medical examinations, all the staff are gathered together by the doctor and invited to describe all the activities that take place in their departments. The reconstruction of the work cycle reveals critical issues that the workers are invited to resolve by creating Participatory Ergonomics Groups (PEGs©) [89]. Each PEG is attended by several doctors (a senior doctor, and two or three trainee doctors), an occupational nurse, and a member of the prevention service. Unlike what occurs in focus groups, in PEGs the occupational doctor does not have an active role in guiding the discussion; he/she simply asks workers to describe their working day and makes sure they all participate in the discussion, regardless of their role in the organization. The information collected during the PEGs is recorded and analyzed using the COREQ criteria [90]; we use the six-step thematic analysis according to Braun and Clarke [91]. After each worker meeting, the doctors review the records describing the work cycle and check them carefully. Next, key aspects of the data and recurring patterns are given specific codes which are subsequently clustered to construct themes and subthemes. In general, themes that can be identified in each department include the architecture of the premises, the number of employees and their training, the organization of shifts, the variation of production flows, and the interaction with other departments or services. The issue of WV frequently arises; workers describe its characteristics and elaborate possible solutions to minimize the problem. In this study, we have not reported the results of the thematic analysis of the various departments, but only anecdotal aspects related to the theme of WV. At the end of the PEG, the doctor communicates the workers’ collective proposals to the company managers.

The third activity involves all workers in the departments where medical surveillance took place and the PEGs were established. The workers are contacted by email and invited to anonymously fill in the complete VIF questionnaire with the description of a violent event that includes the characteristics of the aggressor, the circumstances of the event, their reactions, and the actions taken after the event. Figure 2 summarizes the risk measurement process.

The three activities have different aims: during the medical examination, the aim is to make a census of exposure to violence in the previous year and collect all events important enough for the worker to want to inform the doctor; the creation of worker groups aims to understand how violence interferes with production activities; while the online survey aims to anonymously reveal the characteristics and consequences of violent events.

### 2.3. Statistics

The prevalence of workers who complained of having been exposed to different forms of violence was studied by using the Kolmogorov–Smirnov and Shapiro–Wilk tests to evaluate the type of distribution of the variables and the exact binomial test with the exact Clopper-Pearson 95% Confidence Interval to measure the degree of uncertainty of the distribution. Comparisons between groups were performed using Pearson’s chi-square test. Statistical tests were performed using the IBM SPSS Statistics for Windows, Version 26.0. IBM Corp. Armonk, NY, USA.

## 3. Results

As an example of the results that occupational medicine activities can provide, in this article, we report the observations we made in 2023. Of the 740 workers who were exposed to occupational risks, 651 (88%) participated in the health promotion activities proposed during medical examinations; 473 workers were women (72.7%) and 178 were men (27.3%). The mean age was 49.0 ± 10.3 years (range 23–68). The sample included 111 doctors (17.1%), 300 nurses (46.1%) and 240 other workers (36.9%) such as auxiliary staff, technicians, and clerks. The sample proportions corresponded to those of the entire population.

### 3.1. Data Collected During Medical Examinations

A total of 45 workers (6.9%) reported experiencing one or more physical attacks during the previous year. A total of 83 workers (12.7%) reported having been threatened, while 84 (12.9%) reported harassing behavior. The perpetrators of the attacks were found to be patients (59.9% of cases), visitors (12.2%), colleagues (21.8%), or strangers to the workplace (6.1%).

Women experienced violent behavior more frequently than men, but the difference between genders was significant only for harassment (Table 1).

Doctors and nurses were the categories most exposed to every type of violence. Physical violence was slightly more frequent in nurses than in doctors; however, the difference in other professional categories did not reach statistical significance. The highest frequency of exposure to threats and harassing behavior was reported by doctors. Doctors and nurses were significantly more exposed to verbal violence than other categories of workers (Table 2).

### 3.2. Anecdotal Data Collected in Participatory Ergonomics Groups

In 2023, 9 PEG meetings were held in as many departments of the healthcare company. The WV topic was reported by workers in the description of their activities in all departments. The workers had noted a reduction in all forms of violence during the COVID-19 pandemic, but in 2023, they witnessed an increase in assaults. This topic was present in all the participatory group meetings we conducted in the workplace. Below we include a summary of some of the results related to WV.

In a surgical ward, workers declared they had not experienced any recent episodes of violence. The risk of WV had decreased after the outbreak of the pandemic due to restrictions on visits that allowed only one visitor to be admitted for each patient. Contacts with relatives were supplemented by phone and video calls to nursing staff. Workers said they feared the situation could deteriorate when safety measures put in place during the pandemic were eased.

In a long-term medical ward, according to the workers who participated, verbal aggression towards healthcare personnel on the part of patients’ relatives had decreased after access was limited by procedures introduced during the pandemic. A nurse pointed out that the Region had proposed putting up signs to remind people that aggression is a criminal offence, and that this simple measure had acted as a deterrent. However, when the pandemic ended, a physical attack against a nurse occurred.

In a short-stay surgical ward, workers attributed violence against HCWs to a change in the population’s attitude. Some workers with more experience reported having noticed over the years a change in the reason for visiting relatives: instead of being there to comfort the patient, visitors wanted to test the efficiency of healthcare operations. This created a state of tension among the staff who feared that visitor access would once again become as uncontrolled as it was before the pandemic. The solution suggested by the workers was to limit the number of visitors and control access through security services at the hospital entrance.

In an outpatient department, workers observed that people entered without any screening and crowded together in front of the reception area, sometimes also gaining entry to clinical areas. At times, the resulting confusion gave rise to episodes of verbal aggression. The proposed solution was to create an information point that could filter entry into the healthcare departments.

In the drug addiction assistance service, all categories of staff reported that verbal aggression occurred very frequently and was sometimes accompanied by threats, but no physical aggression had recently been recorded. The staff believed that episodes of violence were “natural” because they were the result of patients’ requests that could not be fulfilled. Patients who were treated in prison usually pressed for treatment because it would guarantee benefits, whereas patients in a state of freedom generally rejected treatment. Although staff are well prepared to deal with these contrasting tensions, the risk cannot be overlooked.

In the Accident and Emergency Department, workers reported that uncivilized and potentially violent behavior on the part of users was especially directed at nurses carrying out the triage because they were the only figures that relatives and visitors came into contact with. Triage involves all nurses in turn; those who have gained greater professional experience are more able to cope with verbal violence than younger ones. The triage structure itself causes problems because there is no area dedicated to receiving relatives and visitors. Long waits, lack of information, and anxiety often make relatives very tense. Moreover, staff have no time to respond to their requests. One of those present during a PEG meeting declared that: “There are so many verbal attacks that if we had to report them all it would be like having a second job”.

### 3.3. Data Collected Through Questionnaires

Workers in the departments where the PEGs were held answered the online VIF questionnaire anonymously. A total of 235 employees were working in the departments inspected, and a total of 163 (71.9%) responded: 49 were males (30.1%), and 114 were females (69.9%). Of the respondents, 31 were physicians, 107 were nurses, and 25 came from other categories of hospital workers (auxiliary staff, technicians, clerks).

The ages of the participants, who were grouped into classes to avoid identification, were comparable to those of the company population whose average age was 49.0 ± 10.3 years; 9.2% were under 30 years of age, 27.6% were aged between 30 and 39, 24.5% between 40 and 49, 30.7% between 50 and 59, and 8.0% were 60 or over.

A considerable percentage of workers reported experiencing violence during the past year, or in previous years (Table 3): 42.9% reported experiencing at least one episode of physical violence, threats, or harassment in the previous year; 55.8% reported that they had experienced at least one episode of violence during their working life. However, only 30 of them reported the details of an episode of violence that referred in more than half of the cases to the current or previous year; in the other cases, the violent episode had occurred years earlier. These attacks were perpetrated mainly by patients (80% of cases) or visitors (16.7%); however, two-thirds of those who had experienced WV failed to indicate the perpetrator.

Males reported physical aggression more frequently than females, but the difference failed to reach statistical significance (Table 4).

Doctors and nurses were attacked more frequently than other healthcare professions (Table 5).

The workers who described an episode of aggression were nearly always nurses; only one doctor provided this information. The episodes were described mainly by workers in outpatient and long-term care services; no workers in the services most at risk (psychiatry and emergency) provided detailed reports.

The attacks were perpetrated by patients in 80% of cases; in the remaining cases, the perpetrators were visitors or relatives of the patients and, very occasionally, strangers to the workplace. Among those who described an episode, the prevalence of females (76.7%) was slightly higher than that of the reference population.

Attacks occurred mainly in the afternoon (46.7% of cases), while the remaining cases of aggression occurred in the morning or during the night shift with equal frequency (26.7%). In some cases, the attackers were under the influence of medication (16.7%), alcohol or drugs (16.7%), or were suffering from cognitive impairment or psychosis (10.0%). The attackers were predominantly male (80%). A total of 63.3% of the perpetrators were aged between 31 and 50 years. Attacks occurred during the administration of treatment (40%), during an interview (33%), or after the patient had made an unanswered request (23.3%). In most cases (63.3%), the attack was unexpected, and the operator was unable to do anything to prevent it. In approximately one-quarter of cases (26.7%), the worker was alone when he/she was attacked

Of the 30 episodes of WV described, the majority were characterized by verbal aggression (26, 86.7%) and incivility (21, 70%), accompanied by punches (4, 13%), kicks (3, 10%), bites (3, 10%), spit (9, 30%), slaps (3, 10%), scratches (3, 10%), torn clothing (2, 7%), hair pulling (1, 3%) and shoving (4, 13%). In 4 cases, a weapon was used.

The attacks resulted in fear (70%), anger (63%), anxiety (40%), anguish (40%), a feeling of defenselessness (40%), humiliation (33%) and disappointment (30%). In 10% of cases, the worker suffered physical harm.

Although the 30 incidents reported in detail by this sample of workers included the most important ones that occurred to each of them, only some of them were reported to superiors (27%), the police (23%), or the worker’s insurance company (7%).

## 4. Discussion

### 4.1. Characteristics of WV

This study describes the method used to monitor WV through health surveillance by taking, as an example, data collected in 2023 from ongoing WV monitoring in a public health company. The variation in WV rates over a 20-year period is the subject of a previous publication [82] and will not be referred to here. We simply indicate that after many years of implementing anti-violence policies, the workers experienced a gradual reduction in WV exposure that peaked in the first phase of the COVID-19 pandemic. The picture changed, however, at the end of the pandemic, when the easing of special measures introduced during the pandemic and a change in public opinion towards the health profession, demonstrated by the emergence of positions contrary to medical practices [92,93,94] and the rise in legal disputes against HCWs [95,96], gradually increased episodes of violence. The year 2023 was the year in which the international authorities officially declared that the COVID-19 pandemic was over [97]. In the same year, the number of HCWs reporting physical or verbal violence returned to pre-pandemic levels. In our study, the percentage of workers who reported having experienced at least one physical assault in the year before their medical examination was 6.9%. This percentage rose to 8.6% among workers interviewed anonymously online. Threats affected a high percentage of workers (12.5% in the survey conducted during medical check-ups, and 25.8% in the online survey).

A risk assessment cannot be based simply on the collection of available data, no matter how carefully this is done. The outcome must be compared with results from other sources. While it is relatively easy to make a historical comparison with data collected in the company by the same method, this is not the case for WV data available in the literature. The first obstacle involves the definition of violence. Since workers have no universal definition of WV [41] and many international bodies have developed their own definitions, authors have adopted different terms for defining WV. It is well known that in the absence of a shared definition, WV is subjective because one employee may perceive “violence” differently from another and interpret it as lower-level antagonism [98]. Numerous examples of unprofessional behavior, including incivility, micro-aggression, harassment, and bullying, are widespread in healthcare and frequently tolerated [99]. The difficulty in defining violence has often troubled researchers, to the extent that some have claimed that WV, like beauty, is in the eye of the beholder [100]. Terminological uncertainty significantly impedes comparisons among various studies, thereby hindering an accurate interpretation of the issue and the formulation of suitable solutions. The method adopted by occupational medicine and used in this study was that of questioning the worker. While methods that collect official reports evaluate only the frequency of major events with serious injuries, the occupational physician can observe the entire pyramid of WV. Only violent behavior that the worker has not yet perceived as such is excluded. From an occupational medicine point of view, WV exists only if the employee perceives it as such. Consequently, we chose to concentrate on the subjective aspect by inquiring whether the worker had experienced violence. Individuals who did not disclose instances of violence during their confidential interviews with the occupational doctor probably perceived these events as irrelevant to their health and work.

Once the definition of WV has been resolved, there still remains the question of how investigations are carried out. Many studies have been conducted on a single category of workers, or on a single department, thus, generating a selection bias [101]. Investigations should include the entire healthcare institution and all categories of workers. The method we propose applies to all workers exposed to occupational risks, regardless of their professional category. Once this has been decided, the next issue concerns the correct method for collecting reports and whether we should wait for the worker to spontaneously report a violent act, perhaps by facilitating his/her task through online or app-based reporting systems [102], or whether we should contact all workers, asking if they have experienced violence in a specific period. A further decision has to be made on whether this census should be open (conducted in person) or anonymous to avoid any reticence. We chose and proposed the last two methods: a face-to-face census of all workers during medical examinations, and an anonymous census through an online questionnaire that gave us the possibility to also count cases in which the worker wishes to remain anonymous. In addition, researchers need to choose the correct questionnaire for collecting responses and also check its validity. The wide variety of available tools hinders comparisons. This aspect is critical in meta-analysis studies that often incorporate surveys conducted with different questionnaires. In our activity we have used the same questionnaire for over 20 years, thus, allowing us to make comparisons over time and between different work environments, or different production activities. The way in which the questionnaire is administered can also influence the result. A generic survey about workplace violence leads respondents to believe that it is desirable to report experiencing violence in order to avoid criticism and increase approval from the researcher [103,104]. We have always chosen not to investigate WV as such, but to include the theme of violence in broader contexts such as health promotion campaigns during medical examinations or the description of work in PEGs after inspections. Only a motivated and thorough method can give a standardized result. Unfortunately, no systematic or meta-analyses have taken all these factors into account or considered the variation in violence rates over time. A critical appraisal of what has been published is urgently needed.

As we have already indicated, detection systems based on the collection of official reports monitor only the most serious cases and consequently estimate prevalence rates of aggression to be in the order of 1–3% [67,68] or much lower [44,66]. Clearly, this is an example of underreporting. The involvement of the occupational physician in medical examinations, inspections, and surveys enables him/her to obtain a complete and up-to-date picture of the true situation.

### 4.2. Strengths and Weaknesses of the Method

Our method of analyzing the WV phenomenon during health surveillance offers several advantages, the first of which concerns costs and sustainability. Our method of collecting data on violence during health surveillance has been in use in the public healthcare company for 20 years: this clearly demonstrates its sustainability. It is important to underline that the monitoring system is entirely financed by the health service, without any additional burden for the employer or for the workers. By asking all the workers to report violent events experienced in the previous year, we minimized underreporting and were able to evaluate the prevalence of assaulted workers in different departments, the distribution of the phenomenon in homogeneous groups, and the evolution of WV over the years. Furthermore, the information collected through the PEGs enabled us to understand the different characteristics that violence assumes in different types of healthcare work. In addition, the collection of anonymous descriptions of violent events allowed us to evaluate their impact. It is important to note that the continuous nature of our survey (questions on violence were inserted into questionnaires used for health promotion activities) helped to prevent overreporting that could have been induced by surveys aimed only at violence. Moreover, involving workers of all categories and departments in reporting the events avoided the problem (frequently found in the literature) of referring to a single professional category and a single type of medical activity. The application of our method does not entail a significant increase in the time required for medical examinations, inspections of the workplace, or other health surveillance obligations. Consequently, the occupational doctor can easily include these activities in his/her health surveillance program and, thus, obtain a much more precise assessment of the risk of violence than is obtainable with other methods.

The main advantage of the method proposed is the accuracy of the risk assessment process. Risk assessment requires a codified series of operations: (i) identification of the hazard (risk factor); (ii) observation or monitoring; iii) measurement of the risk (probability of occurrence) [105,106,107]. Since there are relatively few official reports of violent events, there is an obvious advantage in conducting a census of workers’ experiences of WV. In the absence of data, risk assessment is impossible. In a review of the methods used in Italy to assess the risk of violence in the workplace [80], we documented the prevalence of subjective assessments based on opinions the evaluator had formed from the literature or extemporaneous interviews. These opinions, concerning the probable frequency of events, the severity of the damage they could cause to workers and the effectiveness of possible prevention measures, are used to create algorithms that are not always validated or compared with the real situation. Consequently, they express only an opinion on the potential dangers of a working environment. This kind of procedure, which is far removed from a scientific method, can impede risk prevention. In the absence of official complaints, and on the grounds of an algorithm based on opinions, WV may appear to be under control, even when this is not true. Measuring the risk of violence through workers’ experiences is definitely a method that is more likely to indicate the real situation.

However, the three activities that make up our method have different points of view and provide different measures of risk, as can be seen from a detailed analysis of the characteristics, merits, and defects of each activity. The three questions that are part of the questionnaires administered to workers during periodic medical examinations aim to monitor the prevalence of assaulted workers, not the frequency of assaults. Workers are asked to say whether, in the previous year, they had experienced a physical attack, a threat, or harassment. The one-year period was chosen because it is long enough to encompass working experience, but short enough for the violent event not to be forgotten. In fact, most studies in the literature adopt the duration of one year [108,109,110,111,112]. Since this period also coincides with the usual frequency of medical examinations, the worker can easily remember if the episode occurred before or after the previous check-up.

The questions are designed to establish what the worker perceived as violence. This operational approach overcomes the complex question of establishing exactly what kind of behavior is to be considered violent, because violence becomes what the worker perceives as such. The occupational physician is interested in what the worker intends to communicate, since this report and a subsequent in-depth analysis enable him/her firstly to assess whether support or therapeutic intervention is necessary, and secondly, to determine what actions need to be carried out in the work environment. The doctor is less interested in objectifying the reported episodes of violence, although he/she can do this too after the interview.

This subjective measurement has some limitations: the first is that to ensure that everyone is asked the same questions, it is best to use a written form. For practical reasons, it is preferable to use questionnaires already prepared for this purpose. In 2005, we selected the first three questions from a widely used questionnaire and added them to the forms we use for our annual health promotion campaigns. This procedure ensures that everyone receives the same questions and that the answers are recorded and processed, but it excludes from the survey those who do not wish to participate in health promotion. Although in our experience this part of the workforce is very limited, nevertheless, there is a loss of data. Other workers excluded from the sample are those not exposed to an occupational risk, those who do not undergo medical examinations (e.g., part-time office workers), and those who have been in service for less than one year. Consequently, the annual sample does not exactly correspond to the population under examination, although it aims to get as close to it as possible.

A second limitation is linked to the worker’s decision to make known what happened to him/her. The assumption underlying the request is that if the worker does not report the violent event, he/she does not consider it important enough to be related to the doctor. However, it is also possible that the worker does not want to tell the doctor what happened because he/she fears exposing himself/herself to victimization or thinks the doctor cannot solve the problem [113,114]. This could lead some workers to fail to report experiencing violence, especially if the perpetrator belongs to the same work environment (e.g., a colleague or a superior). The response of victims is greatly influenced by the way in which the occurrence of WV is investigated, anonymously or openly, with or without a guarantee of confidentiality. In fact, fear of victimization and retaliation can be high, especially if the perpetrator has a professional relationship with the victim or has the opportunity to contact him/her outside of work.

Our second monitoring action involves inviting the staff to describe their work and the critical issues they see in it. The description of their experiences enables the physician to ascertain the different characteristics that WV assumes in relation to the healthcare activities that take place in each department. This phase of the evaluation method has some limitations. The main one concerns the number of PEGs the doctor can organize in one year. In a large company, it is usually impossible to cover all the departments. Furthermore, the quality of the data obtained from PEGs varies according to the interaction between workers. Departments with a good work climate provide qualitatively richer, more abundant, and more meaningful data than those in which there is tension between workers. The creation of PEGs, i.e., the involvement of workers in the management of occupational risk, is itself a step towards improving the work environment. Companies that do not encourage participation are not interested in improving the quality of work.

After a period of relative calm, in 2023 there was widespread concern among workers that violence could return to the high rates seen before the pandemic. The observations workers made concerning violence and ways of preventing it are broadly linked to what emerged during other qualitative studies, such as the one conducted in Canada by Brophy et al. [115]. Increased staffing, improved security, personal alarms, new building designs, “zero tolerance” policies, streamlined reporting, utilization of the criminal justice system, improved training, and flagging are just a few of the preventive tactics they proposed. The normalization of violence; underreporting; incivility from patients, visitors, higher-status professionals and supervisors; poor communication; and the fear of retaliation for speaking out publicly, were listed as obstacles along the path to removing hazards. The workers’ distress was exacerbated by inadequate psychological support following an episode of violence.

The third monitoring action involves analyzing the characteristics and consequences of violent episodes. Using an online questionnaire to interview all the workers in the departments where inspections and PEGs were carried out is a kind of verification of the data collected during medical examinations. This phase guarantees worker anonymity that can contribute to eliminating underreporting caused by a possible fear of negative consequences for reporting violent or incorrect behavior.

In fact, in our study, the WV prevalence obtained from this online survey was higher than that observed in interviews with the doctor. However, not all workers take advantage of the opportunity to describe a violent episode in detail. Paradoxically, in the departments that the literature and our own experiences have shown to be at greatest risk (psychiatry and accident and emergency), workers were least likely to extensively describe violent episodes. Since the questionnaires were anonymous, the factor that led to the failure to report violence was probably that the workers in these departments were so frequently exposed to attacks that they considered them an event that did not deserve to be reported, that could not be prevented [116], and was almost a part of their job [117,118]. As one emergency nurse told us, it was not worth wasting time reporting cases!

Overall, our method considerably reduces the underreporting of violent events. Numerous studies have documented the well-known phenomenon of underreporting occupational violence against healthcare personnel. This is considered to be a significant challenge to the proposal of effective preventive measures, since the data used to support them does not accurately reflect the true incidence of all the assaults that occur. It is also difficult to accurately evaluate the efficacy of preventive measures put in place due to underreporting [119], since the latter could frequently be caused by management variables such as the absence of clear adjustments following reporting, a culture that discourages reporting, and the absence of sanctions for offenders. A lack of WV policies, procedures, or training, together with a deficient and problematic reporting mechanism, could be among the organizational problems that enhance underreporting [113].

### 4.3. Indications for Prevention

Our analysis indicated that WV was a very heterogeneous phenomenon within the company, so consequently, prevention measures needed to be differentiated and specific. Violence observed in psychiatric services on the part of patients not in full possession of their mental faculties or under the effect of drugs is not only different from that in emergency departments where the perpetrators are very often relatives or visitors, but also quite different from the unprofessional incivility and bullying that workers sometimes use towards other workers. Although scientific literature gives scarce consideration to the latter type of violence, its importance cannot be overlooked since it has a significant impact on workers’ health. Collecting reports from workers in the PEGs at the end of the inspection helps the occupational physician to correctly frame the type of WV of each department. Statistical analysis of reports collected during medical visits or through online questionnaires will help to quantify the occurrence of different types of events. In every workplace, there are four types of unprofessional behavior: (1) workplace disempowerment, (2) organizational uncertainty, confusion, and stress, (3) social cohesion deficits, and (4) the facilitation of a harmful culture that permits unprofessional behavior [99].

Violence against HCWs is an aspect of a wider issue that stems from the healthcare organization’s inability to identify and address violence and the lack of specific laws to uphold safety standards in the workplace. Inadequate European Union laws for addressing WV in the health profession are partially to blame. HCWs are in need of more legislative commitment to ratify anti-discrimination policies, more institutional support in the form of dedicated funding for targeted interventions, and a chance to voice their needs to decision-makers who can actually alter the status quo [114]. The commitment of the company occupational physician also aims to make workers feel that attention towards WV is high.

What health companies need is not a generic policy against violence, but a series of specific measures for different interventions in different sectors. In psychiatric settings where physical violence is mostly carried out by patients, a study of the patients’ characteristics seems to be the correct approach for effectively minimizing the occurrence of WV and offering better protection for mental sector HCWs [120]. Many researchers believe that educational interventions are effective in improving the ability to deal with situations that lead to violence [121], but others claim that even if education and training can increase awareness of the phenomenon and induce positive attitudes in HCWs, interventions that neglect systemic and organizational issues may be ineffective, especially in the long term [101]. In emergency departments, where a meta-analysis has shown that 77% of staff are exposed to verbal violence, and that family members, other relatives, or friends are the instigators in 75% of cases [122], training designed to recognize aggressive subjects and reduce violence has not proved to be an effective strategy.

Unfortunately, in such complex situations, solutions are decided by the management, and this top-down approach is not always effective. The literature lacks studies based on the collection of staff opinions and alternative bottom-up strategies, although there are signs of some movement in this direction [123]. The method we have been using for many years, by which workers in participatory groups are asked for their opinions on prevention strategies, could provide an indication for hospitals to develop and improve interventions for preventing workplace violence. The data collected during health surveillance have highlighted the urgent need to provide this kind of intervention.

Furthermore, in general terms, preventing the risk of WV cannot be based only on reactive measures such as training workers to recognize the warning signs of violence and defend themselves from aggression, but should include proactive intervention capable of modifying the behavior of potential aggressors. This would be a long-term process, but it should nevertheless be undertaken. Discussion among workers in PEGs can help to increase the level of awareness among workers. Interventions that foster collegiality and team trust among HCWs [124] are appropriate. However, a broader social intervention that changes public opinion’s attitude towards HCWs is desirable. Violent human behavior cannot be eliminated altogether, but throughout human history, the degree of deadly violence has varied in relation to shifts in the socio-political structure of human societies. According to Thomas Hobbes, the social contract was born from the need to limit violence and avoid the mutual extermination of human beings: “homo homini lupus” [125]. Humanity has emerged from all-out war or the repression of the weakest by establishing law. The State is the only legitimate controller of the use of force that may not be exercised by individuals. In the social contract, proactive violence is assigned to the State, which applies it as criminal justice or as war [126]. Other, illegal examples of proactive violence are bullying, stalking, and premeditated homicide. Reactive violence, which is an angry reaction to threats and frustrations, is at the root of WV [127]. Many people believe that WV can be avoided by applying the law. However, the tranquility guaranteed by law is always unstable: it can be broken, and this generates a state of tension that today is known as stress. As Hobbes observed, “The nature of war consists not in actual fighting, but a known disposition thereto, so long as there is no safety from a contrary disposition” [125]. To eliminate violence, it is necessary to move to a higher level of associative life, in which the parties relinquish the use of reactive violence to obtain their rights because the latter would inevitably cause harm to others [128]. Occupational medicine moves precisely in this direction by trying to improve the relationships between workers, patients, visitors, and managers, and consequently the general climate of the working environment [87].

## 5. Conclusions

The occupational physician’s aim is to increase the awareness and participation of workers, which are essential elements for understanding the characteristics and causes of violence and preparing appropriate prevention measures. Attention on the part of management and information dissemination to the community are also vital for achieving shared management of healthcare safety.

In conclusion, measuring the risk of violence through health surveillance activities is a reliable and sustainable method that encourages worker participation and the search for common and decisive solutions.

## Figures and Tables

**Figure 1 ijerph-21-01708-f001:**
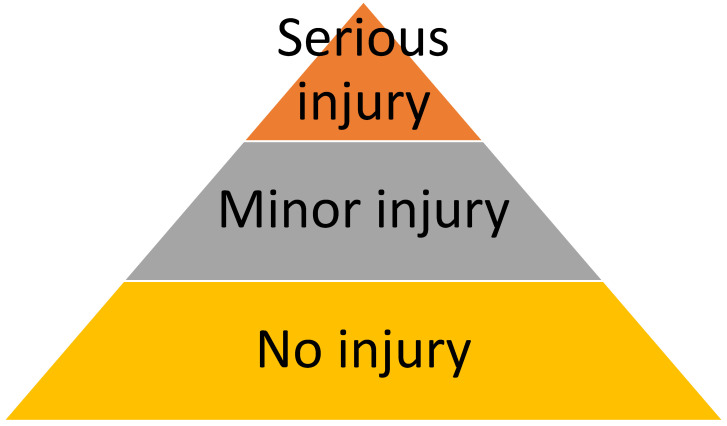
The pyramid of violent events in the workplace.

**Figure 2 ijerph-21-01708-f002:**
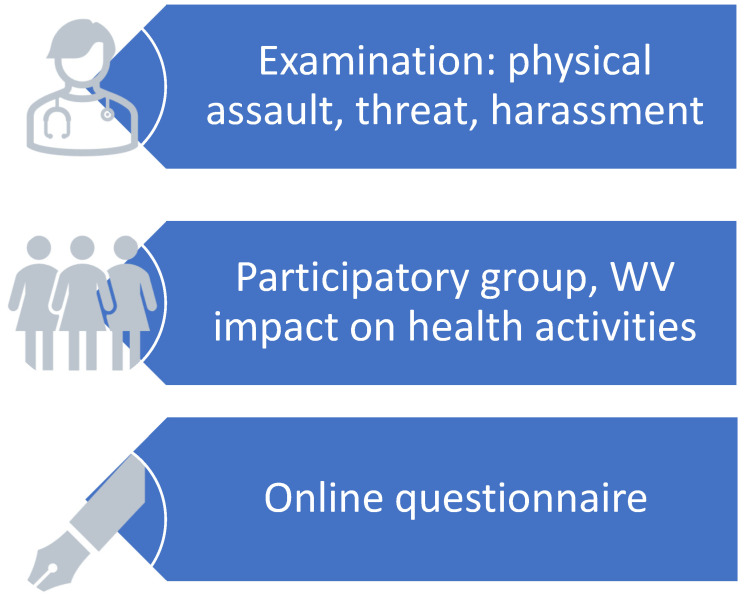
Process for assessing the risk of workplace violence in health surveillance activities.

**Table 1 ijerph-21-01708-t001:** Distribution of cases of violence between the two sexes.

	Male (n = 178)	Female (n = 473)	*p* ^1^
Physical	4.5%	7.8%	0.136
Threat	12.4%	12.9%	0.855
Harassment	8.4%	14.6%	0.037

^1^ Pearson’s chi square.

**Table 2 ijerph-21-01708-t002:** Frequency of different types of violence in professional categories.

	Physician (n = 111)	Nurse (n = 300)	Other ^1^ (n = 240)	*p* ^2^
Physical	8.1%	8.3%	4.6%	0.201
Threat	20.7%	14.7%	6.7%	0.001
Harassment	16.2%	15.3%	8.3%	0.028

^1^ auxiliary staff, technicians, clerks; ^2^ Pearson’s chi square.

**Table 3 ijerph-21-01708-t003:** Prevalence of workers who reported assaults.

Period	Physical Assault	Threat	Harassment
In the past year	8.6%	25.8%	28.8%
In their working life	19.6%	39.9%	37.7%

**Table 4 ijerph-21-01708-t004:** Distribution of cases of violence between the two sexes.

	Male	Female	*p* ^1^
One-year			
Physical assault	7 (14.3%)	7 (6.1%)	0.089
Threat	9 (18.4%)	33 (28.9%)	0.157
Harassment	12 (24.5%)	35 (30.7%)	0.422
Whole working life			
Physical assault	12 (24.5%)	20 (17.5%)	0.306
Threat	18 (36.7%)	47 (41.2%)	0.591
Harassment	18 (36.7%)	37 (32.5)	0.596

^1^ Pearson’s chi square.

**Table 5 ijerph-21-01708-t005:** Frequency of different types of violence in professional categories.

	Physician	Nurse	Other	*p* ^1^
One-year				
Physical assault	2 (6.5%)	12 (11.2%)	0	0.176
Threat	7 (22.6%)	32 (29.9%)	3 (12.0%)	0.165
Harassment	9 (29.0%)	31 (29.0%)	7 (28.0%)	0.995
Whole working life				
Physical assault	5 (16.1%)	27 (25.2%)	0	0.014
Threat	13 (41.9%)	49 (45.8%)	3 (12.0%)	0.008
Harassment	10 (32.3%)	38 (35.5%)	7 (28.0%)	0.760

^1^ Pearson’s chi square.

## Data Availability

Data were deposited on Zenodo DOI 10.5281/zenodo.14260347.

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
