# Peer review of "Measuring the Risk of Violence Through Health Surveillance"

_ijerph, 2024, doi:10.3390/ijerph21121708_

Round 1

Reviewer 1 Report

Comments and Suggestions for Authors

The manuscript is a very interesting and relevant contribution to the existing evidence on workplace violence in the labor sector. However, there are some issues that I believe need to be clarified.

The introduction is extensive, detailed, and comprehensive.

The methodology section is overly brief. It is practically limited to the procedure. There is a lack of information regarding the participants, the variables evaluated, and the data analysis conducted.

If thematic analysis was performed, it would be interesting to clearly outline the themes identified. This could be provided with a chart.

In the results, there should be a clarification regarding the source of the violence (whether it came from colleagues, superiors, or users).

In the discussion, much of the text is dedicated to further addressing aspects related to the methodology employed, and there is little in-depth analysis of the results in comparison to the existing literature. This should be balanced by reducing one section and expanding the other.

Author Response

Reviewer #1

The manuscript is a very interesting and relevant contribution to the existing evidence on workplace violence in the labor sector. However, there are some issues that I believe need to be clarified.

Response: We sincerely thank the reviewer for the appreciation expressed for our work and for the useful advice he gave us to improve the manuscript.

The introduction is extensive, detailed, and comprehensive.

The methodology section is overly brief. It is practically limited to the procedure. There is a lack of information regarding the participants, the variables evaluated, and the data analysis conducted.

If thematic analysis was performed, it would be interesting to clearly outline the themes identified. This could be provided with a chart.

R.: The reviewer rightly observes that we could have given much more information about the exemplary case. This is because this article is mainly aimed at indicating a method that is applicable (and currently applied) in different production sectors, while the case of the healthcare company has been used only as an example. However, we have added the elements that the reviewer indicates, specifying the characteristics of the population used as an example, and detailing the method. In particular on the analysis of PEGs, warning that this study does not report all the qualitative results, but only those referring to WV.

In the results, there should be a clarification regarding the source of the violence (whether it came from colleagues, superiors, or users).

Response: We have gladly added this data.

In the discussion, much of the text is dedicated to further addressing aspects related to the methodology employed, and there is little in-depth analysis of the results in comparison to the existing literature. This should be balanced by reducing one section and expanding the other.

R.: Considering that this article aims to present a risk assessment method, we have given maximum space to the advantages and limitations of the method, comparing the results obtained with this method with those obtained with other methods. See for this the Introduction, from lines 87 to 191. In the second paragraph of the Discussion we have detailed the comparison, explaining, from lines 427 to 486, why there may be differences between the proposed method and other methods in the literature. In the second paragraph of the next subsection (lines 517-534 of the trackchanges version) we have compared the method based on a census of workers with the assessments conducted through algorithms based on opinions; we have already dedicated an article to this topic, to which we refer for further insights.

Reviewer 2 Report

Comments and Suggestions for Authors

Abstract:

- What is the purpose of this study?

- What methods were used to collect and process data in this study?

Keywords:

- Please sort alphabetically

Introduction:

- Please write clearly about the purpose of this study.

Materials and Methods:

- Can you provide more detailed information on how this data collection and processing was carried out?

- Before data collection, did the participants in this study agree to the informed concern? If data collection was online, how were the participants informed of this?

- Has this study received ethical approval from the ethics committee? Please state this in this manuscript

- What application was used to process the data? What statistical tests were used in this study?

Result:

- Lines 256-257: Please explain in more detail what is meant by the previous year here. In 2023? or 2022, a year before data collection?

- Lines 256-261: Why is the data displayed on lines 256-258 different from that in Table 1?

- Lines 257-258: Why is the category "Other violent" not included in table 1?

Conclusions:

- Please do not use references in the conclusions

References:

- Please recheck the writing of the references; for example, in reference number 1, the word "Doi:" uses a capital letter. But in references number 2,3,4, etc., use lowercase letters "doi:"

- The consistency of writing the year varies; for example, the use of the symbol "." before the year in references 13,14,15 even though the other references do not use the symbol "." before the year.

Comments on the Quality of English Language

The quality of the English language may need some improvements in some parts

Author Response

Reviewer #2

Abstract:

- What is the purpose of this study?

R.: We thank the reviewer for this question, which led us to understand that the final section of the Introduction containing the aims of the study was not clearly written. We have rewritten this part. The aim of the article is stated at line 204 as: "Having demonstrated the sustainability, cost-effectiveness and efficiency of the risk assessment method, we aim in this work to indicate its characteristics, advantages and limitations."

- What methods were used to collect and process data in this study?

R.: The method we propose uses three operations that can be carried out by the occupational physician in the workplace. They consist of surveying workers during medical visits, asking if they have suffered violence in the previous year; interviewing workers at the end of the inspection in the workplace to find out if violence interferes with work; questioning workers through an online questionnaire about the characteristics and consequences of violent events of which they have been victims. These methods are analyzed in detail and the material collected in a health company in 2023 is reported as an example of application.

Keywords:

- Please sort alphabetically

R.: Thanks for the tip, we've done it.

Introduction:

- Please write clearly about the purpose of this study.

R.: As above stated, we’ve done it.

Materials and Methods:

- Can you provide more detailed information on how this data collection and processing was carried out?

R.: We have expanded the description of the sample we used as an example, indicating the general characteristics of health surveillance in European countries. We have avoided detailing the health surveillance procedures in the workplace because we believe that the readers, being workers themselves, are aware of them.

- Before data collection, did the participants in this study agree to the informed concern? If data collection was online, how were the participants informed of this?

R.: Although health surveillance activities are mandatory by law and therefore do not require an ethical evaluation, we submitted the research process to the competent ethics committees and obtained informed consent from all workers. The authorizations obtained are reported at the bottom of the article.

- Has this study received ethical approval from the ethics committee? Please state this in this manuscript

R.: We have added this specification: “Although health surveillance activities are mandatory by law and therefore do not require prior authorization, workers undergoing medical examinations are invited to provide consent to the management and electronic processing of their personal data by the occupational physician, even beyond the end of the health surveillance period, for statistical or scientific purposes, in accordance with the Personal Data Protection Code. Furthermore, the research project was approved by the Ethics Committee of the Università Cattolica del Sacro Cuore (protocol code 2896, date of approval 5 December 2019). We obtained informed consent from all workers.

- What application was used to process the data? What statistical tests were used in this study?

R.: We added this indication.

Result:

- Lines 256-257: Please explain in more detail what is meant by the previous year here. In 2023? or 2022, a year before data collection?

R.: The workers were visited between January 1, 2023 and December 31, 2023. The question each of them answered referred to the violence suffered in the previous twelve months, therefore in a period between the date of the visit and the previous year. The Discussion explains that the 12-month period was chosen both because it is the most commonly used in retrospective studies on violence, and because it corresponds to the interval with the previous visit, which is generally 12 months.

- Lines 256-261: Why is the data displayed on lines 256-258 different from that in Table 1?

R.: We thank the reviewer who pointed out a typo. We have checked the other data reported in the tables and confirm them.

- Lines 257-258: Why is the category "Other violent" not included in table 1?

R.: The term "other violent harassment" is pleonastic, we left harassment.

Conclusions:

- Please do not use references in the conclusions.

R.: We accepted the reviewer's suggestion. Furthermore, since the first part of the Conclusions referred to prevention methods, we integrated this part into the Discussion.

References:

- Please recheck the writing of the references; for example, in reference number 1, the word "Doi:" uses a capital letter. But in references number 2,3,4, etc., use lowercase letters "doi:"

R.: We followed the reviewer's advice and corrected the 1st reference.

- The consistency of writing the year varies; for example, the use of the symbol "." before the year in references 13,14,15 even though the other references do not use the symbol "." before the year.

R.: We followed the reviewer's advice. I manually checked 126 references.

Reviewer 3 Report

Comments and Suggestions for Authors

General comments:

I struggled with this article. There were several things that I noted included the fact that the article was too long, went off on too many tangents not related to the study itself, lacked a clear description of the methods, and didn’t really leave the reader with a clear message as to what the results meant. The paper overall appears to be more of a report on a qualitative study of a convenience sample but is not necessarily presented as such.

With all of this said, I believe that this is salvageable but the authors should consider deep cuts to the paper. I think this is an interesting model that I’m sure, is not and could not be done in many places. I think the employees’ views on possible prevention strategies are interesting and are coming as those working on the front line. Again, not an aspect of WV prevention that those of us who research the data have.

I have identified some of the more major issues that I identified in my comments below.

Introduction:

1)      In general, I found the introduction to be much too long and overly text dense. In my opinion, much literature exists regarding WV and could simply be referenced verses being described at length.

2)      The first paragraph seems to be a bit blunt and insensitive. The description of the “phylogenetic component” doesn’t seem to fit with the public health aspect of these types of events. The authors appear to be suggesting that because we’re human beings, WPV is unavoidable (dare I say, preventable?). I agree that it can be prevented but I think there is a better way of describing the human side of WPV. Please revisit the first couple of paragraphs and tone down the rhetoric and stereotyping.

3)      I don’t understand the authors comment about the fact that WV “must be strongly controlled.” While I understand it from a broad perspective, employers/employees cannot control every person coming into a place of business.

4)      Furthermore, the authors suggest that WV is often unacknowledged and poorly studied. I would strongly disagree with both of these comments. WV has been long recognized as an issue and has been widely studied. This was likely not the case prior to the 1980s but at this point, the 80s is now 30-40 years ago.

5)      What is meant or what is included in “personal assistance services” (line 73)? While the paragraph might suggest that healthcare workers are considered personal assistance services, I’m not sure if I would consider these as being equal.

6)      Line 90 – I would argue that a large majority of WV events (those that include bullying, verbal harassment, and other non-physical abuse) is very difficult to identify. I recommend that the authors revisit this sentence.

7)      Related to the above sentence, the next several sentences in this paragraph seem to contradict what the sentence in line 90 suggests. The authors land on the fact that the WV is largely underreported. If that is the case, what is meant by the sentence noted in #5 above?

8)      The paragraph starting at line 86 is really long and difficult to follow. As noted in #9 below, there are some aspects of this paragraph that are contradictory. As the paragraph continues, it isn’t clear who or what risk assessors are and what specific goals the authors are referring to in the last sentence.

9)      Line 103: “….totality of violent events is made up of all the annoying or unpleasant actions that workers perceive as likely to damage their person or the working environment.” This seems a little off putting and could be triggering. WV in general should not be accepted, no matter whether it seems pesky and minimal or whether it results in an injury or death. WV is serious and should be treated as such. However, I feel like the authors’ use of the word “perceives” and the reference to the 2nd part of the triangle as “important” events minimize a whole host of events. I would consider other references such as: verbal and nonphysical abuse not leading to injury, minor injury, and serious injury or death.

10)  I believe the important aspects of the background/introduction really start at line 120. This background regarding WV in other industries in Italy is really what should be highlighted.

11)  The paragraph describing the purpose of the paper (starting at line 185) gets into too much of the methodology of the study and thus is confusing as it lacks context. I recommend revisiting this paragraph and reducing the text to provide a very broad overview of the purpose of the current paper.

Materials and Methods

   1)  Is the health surveillance that the authors described carried out occur in every industry or only healthcare or something else? And where are the medical examinations carried out?

2)      Line 209: The authors reference the occupational physician measuring WV. Is this the only workplace event they measure or do they measure all workplace events?

3)      Do the occupational physicians get any type of training? Especially on how the inspections and surveys should be done? WV is a tough topic to talk about so I suspect that there should be some level of training on the best approaches of asking questions and discussing this topic.

4)      By paragraph 3 of this section, it isn’t clear when the authors refer to “we” (e.g., “we used the six-step thematic analysis), what role the authors play in this study. Are the authors simply using the data that were collected through the three steps or were they part of the team actually doing the health surveillance?

5)      Line 233 – the authors refer to the fact that the question of WV arose frequently. However, earlier in this paragraph, the authors suggest that the occupational physicians don’t ask questions and simply ask the workers to describe their workday. How does these “questions” come up? And who comes up with possible solutions? And how much information is actually communicated to the company managers? Do the workers know up front that information is being shared with the company managers?

6)      Paragraph starting at line 236 – it seems contradictory that the workers were contacted by email and then invited to anonymously complete the questionnaire. Is the questionnaire completed online without any information conveyed regarding who completed it? What is the response rate on this survey?

7)      One thing that the methods don’t discuss is how all of the data from the exams, PEGs, and questionnaires were combined into a dataset. Were these individual data sets? Did the authors have an IRB to conduct the analysis?

Results

1)      How do the proportions of women/men workers compare to the workforce in total?

2)      What occupations were included in the “all other” category reported in table 2?

3)      In the paragraph starting at line 270, the authors suggest that there was an increase in in WV in 2023. Who determined that there was an increase? Was this real or anecdotal?

4)      I recommend the authors consider presenting the data collected through the PEGs in more of an anecdotal approach. Comments in this section felt more subjective and simply represented “fears” for the future and not necessarily centered around the events of the day.

5)      How many workers received the questionnaire (section 3.3)? It’s hard to determine if 163 is a low or high number of respondents.

6)      The authors note that there was a significant percentage of workers that reported experiencing violence. Was this tested? Compared to what? How was it tested? What was the p-value?

7)      Generally, I don’t start sentences with a numeric or decimal value (e.g., see line 328).

8)      Table 4 refers to “whole life.” Is this their entire working life? Does this include only occupational encounters or encounters outside of work as well?

Discussion

1)      The authors note that the variation in WV rates against workers over a 20-year period won’t be referred to but then notes that trend over time in the next sentence. It’s hard to understand to put this comment in perspective without that background information. But it also somehow needs to be related to the current study findings.

2)      The authors discuss the fact that violence is one of victim perception and that events may be defined or viewed differently, that it is subjective. While the authors note that the respondent input was subjective, the authors don’t discuss the impact on their results.

3)      The following paragraph starting at line 411 refers to request modality and poses a serious of questions regarding “reports,” employee reporting, etc. How does any of this paragraph correspond to the current methods and findings? I suggest, unless the authors can identify (and describe) a direct link, this paragraph be deleted or greatly shortened.

4)      The authors spend 2.5 pages describing the strengths and weaknesses. I suggest that the authors review this section and shorten. For example, the authors discuss the three actions that were used to collect information specifically and then note that this activities can easily be inserted into the existing health surveillance program. The actions do not need to be identified specifically here as they were identified earlier. Another example of text that could be deleted or shortened is the following paragraph starting at page 466. It’s not clear how the information presented in this paragraph corresponds to a strength of the study.

5)      I agree with the authors that this type of reporting system appears to be sustainable. However, the results are not generalizable as the cases included are simply those who “volunteered” to participate and do not necessarily represent all workers. While the fact that volunteers were included as a limitation, the authors do not note that results are not generalizable.

6)      The sections of prevention implications and conclusions are both really long and text dense and difficult to follow. Much of the information presented is not clearly connected to the methods or the limited results of the study. The authors add information that is quite tangential and not necessarily additive to the discussion/conclusions.

Comments on the Quality of English Language

As noted in my comments to the editor, I wasn't sure if some of the issues that I noted were issues with a translation to English or simply that the authors write this way in general.

Author Response

Reviewer #3

General comments:

I struggled with this article. There were several things that I noted included the fact that the article was too long, went off on too many tangents not related to the study itself, lacked a clear description of the methods, and didn’t really leave the reader with a clear message as to what the results meant. The paper overall appears to be more of a report on a qualitative study of a convenience sample but is not necessarily presented as such.

With all of this said, I believe that this is salvageable but the authors should consider deep cuts to the paper. I think this is an interesting model that I’m sure, is not and could not be done in many places. I think the employees’ views on possible prevention strategies are interesting and are coming as those working on the front line. Again, not an aspect of WV prevention that those of us who research the data have.

I have identified some of the more major issues that I identified in my comments below.

Response: We are especially grateful to the reviewer for the complex series of questions he/she raised, in the genuine and fundamental task of contributing to the improvement of the article. As the chief of a section of this journal, I know how difficult it is to find reviewers who apply themselves to this difficult task. For this reason, we have tried to put into practice all the advice we have received, with the certainty that these changes will increase the usability of the study by readers of all backgrounds.

As the reviewer correctly pointed out, this study is the proposal of a method for assessing the risk of violence in the workplace. We are proud that he/she appreciated the proposal of an original model and understood that it is not applied in many places. This does not mean that it is not applicable everywhere. Our university team has applied this method in all companies and in all work sectors. To exemplify the application of the method, in this article the case of a health company has been exposed. Some reports on violence in this and other companies followed by our team have been published in the past, but they are not the subject of this article.

Introduction:

1)      In general, I found the introduction to be much too long and overly text dense. In my opinion, much literature exists regarding WV and could simply be referenced verses being described at length.

R.: I am personally very inclined to follow the reviewer's advice, because I prefer a very short introduction. However, not all reviewers and editors have the same opinion. Reviewer #1 praised us because "The introduction is extensive, detailed, and comprehensive." So, we tried to take this advice to heart, without distorting the manuscript. We have eliminated over 200 words from the Introduction and summarized the concepts expressed in various parts of the manuscript,

2)      The first paragraph seems to be a bit blunt and insensitive. The description of the “phylogenetic component” doesn’t seem to fit with the public health aspect of these types of events. The authors appear to be suggesting that because we’re human beings, WPV is unavoidable (dare I say, preventable?). I agree that it can be prevented but I think there is a better way of describing the human side of WPV. Please revisit the first couple of paragraphs and tone down the rhetoric and stereotyping.

R.: Following the advice that the reviewer gave us, the first paragraph reports in a very synthetic way the results of the scientific literature, without describing them at length. The 12 bibliographic entries of the 1st paragraph refer to genetic, neuropsychiatric, behavioral, immunological and metabolic studies that constitute the basis of the problem and are not extraneous to public health.

3)      I don’t understand the authors comment about the fact that WV “must be strongly controlled.” While I understand it from a broad perspective, employers/employees cannot control every person coming into a place of business.

R.: The term "risk control" is commonly used in risk management and accident prevention. We have replaced "strongly controlled" with "thoroughly prevented", to avoid misunderstandings by those unfamiliar with risk control strategies. Nowhere in the manuscript does it say that employers/employees must control every person coming into a place of business.

4)      Furthermore, the authors suggest that WV is often unacknowledged and poorly studied. I would strongly disagree with both of these comments. WV has been long recognized as an issue and has been widely studied. This was likely not the case prior to the 1980s but at this point, the 80s is now 30-40 years ago.

R.: We know that there are many studies on WV, but not all of them have the necessary quality. For this reason, we are proposing a method. Many authors have noted that WV is understudied. Ma and Thomas affirmed that “Workplace violence in the healthcare system is vastly understudied” [ref. 38 in the previous version of the manuscript]. Schmidt et al. affirmed that “Workplace violence against neurosurgeons is an understudied phenomenon”. Jaber et al. stated that “violence in nursing workplace is still understudied”. Alharbi et al. affirmed that “Workplace violence is a common problem that is encountered by healthcare workers worldwide; however, it is still under-studied”. Zelnick et al. wrote that “Workplace violence is a serious and surprisingly understudied occupational hazard in social service settings.”. We have indicated the references.

5)      What is meant or what is included in “personal assistance services” (line 73)? While the paragraph might suggest that healthcare workers are considered personal assistance services, I’m not sure if I would consider these as being equal.

R.: We specified "healthcare and personal assistance services".

6)      Line 90 – I would argue that a large majority of WV events (those that include bullying, verbal harassment, and other non-physical abuse) is very difficult to identify. I recommend that the authors revisit this sentence.

R.: We could not understand what the reviewer was referring to. In the paragraph we have reported the risk assessment procedure which takes place through the phases of identification, measurement, monitoring and auditing [lines 87-91]. On line 91, based on what we have said since the beginning of the article, we stated that the WV hazard is easily identifiable in all work environments. Our sentence [“Information gathered on violence leads us to believe that WV hazard can be easily identified in all work environments”] is clear enough. The above sentence [“a large majority of WV events (those that include bullying, verbal harassment, and other non-physical abuse) is very difficult to identify”] that the reviewer attributes to us, is nowhere to be found in the manuscript.

7)      Related to the above sentence, the next several sentences in this paragraph seem to contradict what the sentence in line 90 suggests. The authors land on the fact that the WV is largely underreported. If that is the case, what is meant by the sentence noted in #5 above?

R.: As mentioned above, we can't understand. From line 92 to 101 we reported 8 studies on the problem of underreporting. As is known, the identification of a hazard or risk factor is not equivalent to the measurement of the probability i.e. the extent of the risk. There is no contradiction for those familiar with risk assessment procedures. To move from identifying a risk to measuring its probability, events must be detected.

8)      The paragraph starting at line 86 is really long and difficult to follow. As noted in #9 below, there are some aspects of this paragraph that are contradictory. As the paragraph continues, it isn’t clear who or what risk assessors are and what specific goals the authors are referring to in the last sentence.

R.: Taking into account the reviewer's useful suggestions expressed in the following point, we have modified the structure of the paragraph, the definitions and Figure 1. As mentioned above, paragraph lines 87 to 103 describe the risk management cycle and highlight the problem of underreporting. From lines 105 to 124 we describe the different types of violent events and explain how each data collection method is suitable to collect a part of these events.

The risk assessor by law is the employer, or the person delegated by him/her (who in Europe is called the Prevention and Protection Service Manager) and the occupational health physician charged of the medical surveillance of workers who collaborates with him/her. The employer has the duty to assess all professional risks and implement the necessary prevention measures. We have not added this information because it is universally known to those involved in health and safety in the workplace.

9)      Line 103: “….totality of violent events is made up of all the annoying or unpleasant actions that workers perceive as likely to damage their person or the working environment.” This seems a little off putting and could be triggering. WV in general should not be accepted, no matter whether it seems pesky and minimal or whether it results in an injury or death. WV is serious and should be treated as such. However, I feel like the authors’ use of the word “perceives” and the reference to the 2nd part of the triangle as “important” events minimize a whole host of events. I would consider other references such as: verbal and nonphysical abuse not leading to injury, minor injury, and serious injury or death.

R.: We thank the reviewer who provided the exact terms to refer to. We have modified figure 1 and the corresponding text to introduce the suggested definitions. We are talking about events that workers perceive as violent. If a behavior is not perceived as violent, there is no chance that it will be reported or detected. In the new version of the paragraph, we have clarified the meaning.

10)  I believe the important aspects of the background/introduction really start at line 120. This background regarding WV in other industries in Italy is really what should be highlighted.

R.: We thank the reviewer who understood how the previous long exposition, useful for readers who are not familiar with the procedures of risk assessment and prevention of health and safety risks but boring for others, was necessary to make it clear that with some methods it is possible to measure only the vertex of the triangle of violent events, while with other methods it is possible to measure all the violence that workers have suffered.

11)  The paragraph describing the purpose of the paper (starting at line 185) gets into too much of the methodology of the study and thus is confusing as it lacks context. I recommend revisiting this paragraph and reducing the text to provide a very broad overview of the purpose of the current paper.

R.: We welcomed the reviewer's suggestion very gladly. The purpose of the work - proposal of a WV risk assessment method - is now defined by a single sentence. The use of data obtained from the observation of a company is clearly indicated as an example.

Materials and Methods

  • Is the health surveillance that the authors described carried out occur in every industry or only healthcare or something else? And where are the medical examinations carried out?

R.: We already indicated that “workers exposed to occupational risks are subjected to health surveillance”, whatever their work sector (agriculture, industry, commerce, services). In the new version of the manuscript, we have added this information. The visits are generally made in a local of the same company, but this is not an obligation.

  • Line 209: The authors reference the occupational physician measuring WV. Is this the only workplace event they measure or do they measure all workplace events?

R.: Health surveillance allows the physician to monitor many variables of occupational interest, such as work ability, work-related stress, physical and mental health status, and sleep quality. In this article, we have only referred to violence.

  • Do the occupational physicians get any type of training? Especially on how the inspections and surveys should be done? WV is a tough topic to talk about so I suspect that there should be some level of training on the best approaches of asking questions and discussing this topic.

R.: The reviewer is absolutely right. The occupational physician needs specific training to learn how to correctly measure violence and the other variables I mentioned above. In the company we used as an example, two of the authors were occupational physicians, the other two were doctors in training in occupational health. The university had chosen this health company as a suitable structure for specialist training.

  • By paragraph 3 of this section, it isn’t clear when the authors refer to “we” (e.g., “we used the six-step thematic analysis), what role the authors play in this study. Are the authors simply using the data that were collected through the three steps or were they part of the team actually doing the health surveillance?

R.: All authors of the article were part of the surveillance team, two as specialists and two as interns. We have added this specification in the methods section.

  • Line 233 – the authors refer to the fact that the question of WV arose frequently. However, earlier in this paragraph, the authors suggest that the occupational physicians don’t ask questions and simply ask the workers to describe their workday. How does these “questions” come up? And who comes up with possible solutions? And how much information is actually communicated to the company managers? Do the workers know up front that information is being shared with the company managers?

R.: In the new version of the manuscript we have added some information on the PEGs and on the methods of thematic analysis of the results of the meetings with the workers. A detailed analysis of the PEG methodology was beyond the scope of this work; however, we have included some bibliographical entries useful for further study.

In the PEG, workers describe their work and the difficulties they encounter. One difficulty could be the WV. The reviewer correctly pointed out that the term "question" is ambiguous, we should have used the term "issue". We corrected the text.

The solutions are sought up and discussed by the workers. The doctor simply records them and transmits them to management as workers' proposals. The workers know that all occupational health activities are subject to professional secrecy (Hippocratic Oath) and confidentiality, so the transmission from the doctor to the employer, to safety managers and to workers' representatives can only take place in a collective anonymous form.

  • Paragraph starting at line 236 – it seems contradictory that the workers were contacted by email and then invited to anonymously complete the questionnaire. Is the questionnaire completed online without any information conveyed regarding who completed it? What is the response rate on this survey?
  1. The VIF questionnaire was created to report on a violent event; it investigates the characteristics of the aggressor, the circumstances of the event and its consequences. The questionnaire required the respondent's gender, age group and job title. We sent the questionnaires only to those who worked in the departments in which we established the PEGs. The reviewer correctly reported that we had forgotten to indicate the response rate (72%). We have added this data in the new version.

  • One thing that the methods don’t discuss is how all of the data from the exams, PEGs, and questionnaires were combined into a dataset. Were these individual data sets? Did the authors have an IRB to conduct the analysis?

R.: All workers have an individual risk file, which contains health data and data relating to occupational risks. Individual data are combined into homogeneous groups by the doctor and transmitted in collective anonymous form to the employer, the person responsible for the prevention service and the workers' representatives. We don't really understand what IRB is. If it's an ethics committee, the study had the required authorizations. If your request is about data analysis, we have included some information about the statistical method.

Results

  • How do the proportions of women/men workers compare to the workforce in total?

R.: The sample proportions corresponded to those of the entire population. We added this specification to the first line of the results.

  • What occupations were included in the “all other” category reported in table 2?

R.: We had already explained in the text [line 253] that they were auxiliary staff, technicians, clerks. We had added a note to the table.

  • In the paragraph starting at line 270, the authors suggest that there was an increase in in WV in 2023. Who determined that there was an increase? Was this real or anecdotal?

R.: The text is very clear. Workers had noted a reduction in all forms of violence during the COVID-19 pandemic, but in 2023 they witnessed an increase in assaults. The data is real and was published in the article referenced. The sharp increase in WV after the pandemic observed in this company corresponds to a phenomenon that has occurred on a global scale and on which there is a large literature. We have cited some studies in this regard.

  • I recommend the authors consider presenting the data collected through the PEGs in more of an anecdotal approach. Comments in this section felt more subjective and simply represented “fears” for the future and not necessarily centered around the events of the day.

R.: As clearly explained, PEGs do not focus on violence, but on the complex of productive activities. The topic of violence is only one of the problems highlighted by the workers. The doctor does not lead the discussion as in a focus group, but simply records what is reported. The workers' experiences and their suggestions are necessarily anecdotal and are not sufficient to describe the phenomenon as a whole; however, they are useful contributions from workers that the doctor can connect with the information he/she receives during his surveillance role.

  • How many workers received the questionnaire (section 3.3)? It’s hard to determine if 163 is a low or high number of respondents.

R.: The reviewer is right, we had failed to indicate the total number and the participation rate; we have corrected it.

  • The authors note that there was a significant percentage of workers that reported experiencing violence. Was this tested? Compared to what? How was it tested? What was the p-value?

R.: The reviewer is right. We were wrong to use the term "significant" which has a precise statistical value. We have replaced it with "consistent".

  • Generally, I don’t start sentences with a numeric or decimal value (e.g., see line 328).

R.: We have solved the problem reported by the reviewer by changing the punctuation.

  • Table 4 refers to “whole life.” Is this their entire working life? Does this include only occupational encounters or encounters outside of work as well?

R.: As in Table 3, also in Tables 4 and 5 the workers were questioned about their entire working life. We have corrected the tables.

Discussion

  • The authors note that the variation in WV rates against workers over a 20-year period won’t be referred to but then notes that trend over time in the next sentence. It’s hard to understand to put this comment in perspective without that background information. But it also somehow needs to be related to the current study findings.

R.: As the reviewer correctly noted, we have reported only one of the data that emerged from the company's observation to allow the reader to interpret the workers' responses.

  • The authors discuss the fact that violence is one of victim perception and that events may be defined or viewed differently, that it is subjective. While the authors note that the respondent input was subjective, the authors don’t discuss the impact on their results.

R.: We thank the reviewer because he/she gave us the opportunity to complete the paragraph on the importance of workers' perception of WV with a comparison between the method proposed here and those based on counting reports to the insurance company or to the authorities. While the latter only evaluate the frequency of major events with serious injuries, the occupational physician can observe the entire pyramid of events. Only violent behaviors that the worker has not yet perceived as such are excluded.

  • The following paragraph starting at line 411 refers to request modality and poses a serious of questions regarding “reports,” employee reporting, etc. How does any of this paragraph correspond to the current methods and findings? I suggest, unless the authors can identify (and describe) a direct link, this paragraph be deleted or greatly shortened.

R.: We deeply thank the reviewer for giving us the opportunity to explain the reasons for the methodological choices we propose. The main problems of epidemiological design, choice of instrument and administration method were addressed many years ago, when we developed the method proposed here.

  • The authors spend 2.5 pages describing the strengths and weaknesses. I suggest that the authors review this section and shorten. For example, the authors discuss the three actions that were used to collect information specifically and then note that this activities can easily be inserted into the existing health surveillance program. The actions do not need to be identified specifically here as they were identified earlier. Another example of text that could be deleted or shortened is the following paragraph starting at page 466. It’s not clear how the information presented in this paragraph corresponds to a strength of the study.

R.: Accepting the reviewer's suggestion, we cut the paragraph that summarized the proposed methods. We underlined the advantage of making a census of workers that contrasts with the common and deteriorated use of algorithms not based on observations but on hypotheses.

  • I agree with the authors that this type of reporting system appears to be sustainable. However, the results are not generalizable as the cases included are simply those who “volunteered” to participate and do not necessarily represent all workers. While the fact that volunteers were included as a limitation, the authors do not note that results are not generalizable.

R.: The reviewer attributes to us a term and a setting that we have not used. Health surveillance is not voluntary, but mandatory. Workers can refuse to participate in health promotion activities, but in doing so they exercise their informed consent and only a small percentage of workers do not participate. In the example we have illustrated, 88% of workers participated in the census during visits and 72% responded to the online questionnaire. We do not believe that this type of response should be overlooked. The data loss, which we have reported, is minimal and does not affect the generalizability of the experience.

Of course, the worker who has suffered violence can avoid reporting it. No one can measure the data that is kept secret, so to evaluate the risk it is necessary to report the events. The method we propose minimizes underreporting compared to what happens in the examples we cited of two structures in the same Lazio region in which it turns out that only 2 workers out of a thousand suffer violence. Our method is certainly better than the algorithms based on impressions, opinions or beliefs of the evaluator that are widely used.

  • The sections of prevention implications and conclusions are both really long and text dense and difficult to follow. Much of the information presented is not clearly connected to the methods or the limited results of the study. The authors add information that is quite tangential and not necessarily additive to the discussion/conclusions.

R.: We agree with the reviewer that in some cases the discussion may be difficult to follow for readers who are not familiar with the risk assessment process. We have tried to make clear the logical steps that closely connect the text with the need to explain a method. The Discussion is very complex and for this reason it has been divided into sub-sections. The part that reports the measures to prevent violence has been remodeled, including the distinction between reactive and proactive measures and part of the text that was previously at the end of the article. The conclusions are now very brief and exclusively addressed to the method presented.

Comments on the Quality of English Language

As noted in my comments to the editor, I wasn't sure if some of the issues that I noted were issues with a translation to English or simply that the authors write this way in general.

R.: We would like to thank the reviewer who had to examine a study on risk assessment methodology, whose language is necessarily very technical and refers to the knowledge of a series of health and safety standards and procedures.

Round 2

Reviewer 1 Report

Comments and Suggestions for Authors

After the modifications made, I consider that the manuscript is suitable for publication.

Author Response

Reviewer #1

After the modifications made, I consider that the manuscript is suitable for publication.

Response: We thank the reviewer for the advice that allowed us to improve the manuscript.

Reviewer 3 Report

Comments and Suggestions for Authors

General feedback: I commend the authors for carefully considering the reviewers’ comments and making modifications as necessary. I believe that the modifications are moving the manuscript in the correct direction but as indicated below, I think there are still some changes that the authors should consider.

Introduction:

1)      I personally still feel that the introduction is too long. However, I recognize the author’s note regarding other’s feeling like the additional information is useful. I think if the journal is amenable to this, I am fine with the introduction as is.

2)      Page 2, line 47 (“…..WV is quantified not only by the days of sickness absence of HCWs (fortunately a limited number….”): I think there is something missing between “sickness” and “absence of HCWs.” Please review and modify accordingly.

3)      Page 2, line 69 (“…..can be further categorized into Type III-A (horizontal) and Type III-B (vertical).”): I’m presuming that horizontal is co-worker vs co-worker and vertical is supervisor vs subordinate. It took me a minute to realize this. I suggest not including this additional information as it might need a bit more explanation.

4)      Page 2, line 74 (“….Type III aggression on the part of colleagues or other types of violence cannot be excluded.”): Consider simplifying this to “….Type III violence cannot be excluded.” Or “…..Type III violence can also occur.”

5)      Page 2, line 78 (“WV in healthcare is particularly appalling….”): I personally am not crazy about the use of the word appalling here. Maybe consider another word. I also don’t believe that this sentence takes into account that violence may actually involve the patient who may not always be so fragile or vulnerable.

6)      Page 3, line 104 (“WV can be illustrated as a pyramid (Figure 1) made up of all the annoying or unpleasant actions that workers perceive as likely to damage their person or the working environment.”): I still feel that this sentence is a bit off putting and/or triggering. I strongly encourage the authors to consider just ending the sentence after “(Figure 1).” And then delete the remainder of this sentence. I would then also delete the 2nd sentence. The second sentence really doesn’t address the pyramid itself.

7)      Page 3, paragraph staring at line 104: As I re-read this paragraph, I realize that the information provided in this paragraph is a mix of describing the pyramid and whether or not cases are reported. I strongly encourage the authors to tease out the information specifically regarding pyramid and determine if the information regarding reporting (or not reporting) can be added later (or if it is already stated elsewhere). Furthermore, is the pyramid something that the authors created themselves or can this be referenced?

8)      Page 3, starting at line 125: this is one area, per my above comment, where I think text could be cut. While I understand the importance of discussing/describing the various methods of collecting data, some of the more detailed information regarding these methods could be minimized.

Materials and Methods

1)      Page 5, line 220 (“…..period, By signing the personal health document, the workers also agreed to collective anonymous publication….”): I believe there should be a period after the word period. And I don’t quite understand what is meant by “collective anonymous publication….”

2)      Page 5, line 227 (“The prevalence of workers who compla……”): This paragraph seems out of place. Finish discussing the actual data collection process first and then describe how the data were analyzed. In fact, the authors could go into a bit more detail about how the data were assembled (e.g., were the data added to an Excel table and analyzed using Excel or something else).

Results

1)      Page 7, line 304: Since you noted that anecdotal information only would be presented from the PEG, I would retitle to something like Anecdotal Examples from Participatory Ergonomics Groups.

Discussion

1)      I still think that the lengthy description of the strengths and weaknesses of this study is too long. I strongly encourage the authors consider describing the strengths and limitations at a higher level and avoid getting too far into the weeds. Given that the authors main goal is highlighting this methodology, I suggest that the authors focus on the methodology and not on the data obtained. 1)       

2)      Furthermore, I suggest refocusing section 4.3 on the use of the METHODS that were used. Again, the authors get too far into the weeds with discussing types of unprofessional behavior, organization’s inability to identify and address WV, etc. What are the indications for prevention for the health surveillance tool, the PEG, and the questionnaire?

Author Response

Reviewer #3

General feedback: I commend the authors for carefully considering the reviewers’ comments and making modifications as necessary. I believe that the modifications are moving the manuscript in the correct direction but as indicated below, I think there are still some changes that the authors should consider.

Response: We sincerely thank the reviewer for the commitment with which he/she reviewed our study and for the useful indications he/she provided us.

Introduction:

  • I personally still feel that the introduction is too long. However, I recognize the author’s note regarding other’s feeling like the additional information is useful. I think if the journal is amenable to this, I am fine with the introduction as is.

Response: We thank you for your advice. Even in this second revision we have tried to summarize the text as much as possible, without sacrificing clarity. We were happy to remove all the parts of the Introduction that the reviewer pointed out to us.

Personally, I totally agree that a short text is better than a too long one; the Introduction in my articles generally does not exceed 1000 words and the text as a whole is very synthetic, so much so that I have been invited several times by the editorial staff to increase the length of the manuscript. This article has the peculiarity of bringing together a description of the method with an emblematic case and this pushes towards doubling the text, especially in the Introduction which must explain the reasons of interest of the work.

  • Page 2, line 47 (“…..WV is quantified not only by the days of sickness absence of HCWs (fortunately a limited number….”): I think there is something missing between “sickness” and “absence of HCWs.” Please review and modify accordingly.

Response: In truth, there was no loss of words. Sick days due to violence are low compared to total employee absenteeism, which has led to underestimate the problem of violence. We have changed the text, so that it is now more understandable. Now it is as follows: “Only in the mid-1990s did it become clear that the economic damage caused by WV is quantified not only by the number of sick days suffered by the HCWs (fortunately a limited number) who suffer the greatest physical damage, but also by the considerable emotional and motivational effects of the various forms of violence [25].”

  • Page 2, line 69 (“…..can be further categorized into Type III-A (horizontal) and Type III-B (vertical).”): I’m presuming that horizontal is co-worker vs co-worker and vertical is supervisor vs subordinate. It took me a minute to realize this. I suggest not including this additional information as it might need a bit more explanation.

Response: We agree with the reviewer. We cut this part.

  • Page 2, line 74 (“….Type III aggression on the part of colleagues or other types of violence cannot be excluded.”): Consider simplifying this to “….Type III violence cannot be excluded.” Or “…..Type III violence can also occur.”

R.: We agree. We changed according to suggestion: “other types of violence cannot be excluded”

  • Page 2, line 78 (“WV in healthcare is particularly appalling….”): I personally am not crazy about the use of the word appalling here. Maybe consider another word. I also don’t believe that this sentence takes into account that violence may actually involve the patient who may not always be so fragile or vulnerable.

R.: The reviewer is right. The term "appalling" is inappropriate. In the previous version, we had used the term "execrable" here. Furthermore, the reviewer correctly points out that the patient can be not only the indirect victim of violence, but also the perpetrator. In conclusion, we have used the adjective "serious" and removed the phrase about the patient's fragility.

  • Page 3, line 104 (“WV can be illustrated as a pyramid (Figure 1) made up of all the annoying or unpleasant actions that workers perceive as likely to damage their person or the working environment.”): I still feel that this sentence is a bit off putting and/or triggering. I strongly encourage the authors to consider just ending the sentence after “(Figure 1).” And then delete the remainder of this sentence. I would then also delete the 2nd The second sentence really doesn’t address the pyramid itself.

R.: We thank the reviewer, who identified a portion of text that can be eliminated, without making the text less understandable, but rather increasing readability. We have cut the following sentences: “made up of all the annoying or unpleasant actions that workers perceive as likely to damage their person or the working environment. If the behavior is not perceived as violent, it will not be reported or detected.”

  • Page 3, paragraph staring at line 104: As I re-read this paragraph, I realize that the information provided in this paragraph is a mix of describing the pyramid and whether or not cases are reported. I strongly encourage the authors to tease out the information specifically regarding pyramid and determine if the information regarding reporting (or not reporting) can be added later (or if it is already stated elsewhere). Furthermore, is the pyramid something that the authors created themselves or can this be referenced?

R.: We gladly accepted the suggestion. We separated the description of the pyramid of events from that of the methodologies. The pyramid is a graphic device that we used to explain the concept, it has not been used by others and there are no references.

  • Page 3, starting at line 125: this is one area, per my above comment, where I think text could be cut. While I understand the importance of discussing/describing the various methods of collecting data, some of the more detailed information regarding these methods could be minimized.

R.: Following the reviewer's suggestion, we cut this part of the Introduction

Materials and Methods

  • Page 5, line 220 (“…..period, By signing the personal health document, the workers also agreed to collective anonymous publication….”): I believe there should be a period after the word period. And I don’t quite understand what is meant by “collective anonymous publication….”

R.: The reviewer reported a typo. The comma after the word period was a period. Publication in anonymous collective form is the legally permissible method of disseminating the results of health surveillance

  • Page 5, line 227 (“The prevalence of workers who compla……”): This paragraph seems out of place. Finish discussing the actual data collection process first and then describe how the data were analyzed. In fact, the authors could go into a bit more detail about how the data were assembled (e.g., were the data added to an Excel table and analyzed using Excel or something else).

R.: Following the reviewer's suggestion, we have divided the chapter into three sections, Population, Measures, and Statistics. The indicated part has been moved to section 2.3. Statistics.

Results

  • Page 7, line 304: Since you noted that anecdotal information only would be presented from the PEG, I would retitle to something like Anecdotal Examples from Participatory Ergonomics Groups.

R.: We gladly accepted the reviewer's idea, changing the title of the section.

Discussion

  • I still think that the lengthy description of the strengths and weaknesses of this study is too long. I strongly encourage the authors consider describing the strengths and limitations at a higher level and avoid getting too far into the weeds. Given that the authors main goal is highlighting this methodology, I suggest that the authors focus on the methodology and not on the data obtained. 1)       

R.: We absolutely agree on the need to indicate the advantages and limitations of the method regardless of the case that is given here as an example. In this section, in fact, we have analyzed the pros and cons of the method, also considering the experiences collected in other companies in the literature.

  • Furthermore, I suggest refocusing section 4.3 on the use of the METHODS that were used. Again, the authors get too far into the weeds with discussing types of unprofessional behavior, organization’s inability to identify and address WV, etc. What are the indications for prevention for the health surveillance tool, the PEG, and the questionnaire?

R.: We welcomed the reviewer's suggestion, which gave us the opportunity to explain how the proposed method supports the prevention prospects. We have added a few short sentences to help the reader understand this connection.